# Quality of Reporting of Adverse Drug Reactions to Antimicrobials Improved Following Operational Research: A before-and-after Study in Sierra Leone (2017–2023)

**DOI:** 10.3390/tropicalmed8100470

**Published:** 2023-10-09

**Authors:** Thomas A. Conteh, Fawzi Thomas, Onome T. Abiri, James P. Komeh, Abdulai Kanu, Joseph Sam Kanu, Bobson Derrick Fofanah, Pruthu Thekkur, Rony Zachariah

**Affiliations:** 1National Pharmacovigilance Center, Pharmacy Board of Sierra Leone, Ministry of Health, Freetown 047235, Sierra Leone; fthomas@pharmacyboard.gov.sl (F.T.); otabiri@pharmacyboard.gov.sl (O.T.A.); kompjames@yahoo.com (J.P.K.); akanu248@gmail.com (A.K.); 2College of Medicine & Allied Health Sciences, University of Sierra Leone, Freetown 047235, Sierra Leone; 3National Disease Surveillance Program, Directorate of Health Security and Emergencies, Ministry of Health and Sanitation, Freetown 047235, Sierra Leone; josephsamkanu@gmail.com; 4World Health Organization, Country Office, Freetown 047235, Sierra Leone; fofanahb@who.int; 5Centre for Operational Research, International Union against Tuberculosis and Lung Disease, 75001 Paris, France; pruthu.tk@theunion.org; 6UNICEF, UNDP, World Bank, WHO Special Programme for Research and Training in Tropical Diseases (TDR), 1211 Geneva, Switzerland; zachariahr@who.int

**Keywords:** SORT IT, operational research, VigiBase, drug safety, health systems strengthening

## Abstract

**Background:** The quality of pharmacovigilance data is important for guiding medicine safety and clinical practice. In baseline and follow-up studies after introducing interventions to improve the quality of reporting of Individual Case Safety Reports (ICSRs) in Sierra Leone, we compared (a) timeliness and completeness of reporting and (b) patient outcomes classified as ‘recovering’. **Methods**: Baseline (January 2017–December 2021) and follow-up (June 2022–April 2023) studies of ICSRs in the national pharmacovigilance database. Interventions introduced following recommendations from the baseline study included: updating standard operating procedures and guidelines, setting performance targets follow-up of patient outcomes, and training. **Results:** There were 566 ICSRs in the baseline study and 59 in the follow-up study. Timelines (reporting < 30 days) improved by five-fold (10% at baseline to 47% in follow-up). For the completeness of variables in ICSRs (desired threshold ≥ 90%),this was 44% at baseline and increased to 80% in the follow-up study. ‘Recovering’ outcomes reduced from 36% (baseline study) to 3% (follow-up study, *p* < 0.001). **Conclusions:** Significant improvements in timeliness, completeness, and validation of ICSRs were observed following operational research in Sierra Leone. While enhancing pharmacovigilance and patient safety, this study highlights the important synergistic role operational research can play in improving monitoring and evaluation systems.

## 1. Introduction

An adverse drug reaction (ADR) is defined as “any response to a drug which is noxious and unintended and that occurs at doses normally used in human beings for prophylaxis, diagnosis, therapy of disease, or for the modification of physiological functions’’ [1]. Vigilant monitoring and reporting of ADR data would have major benefits for the clinical management of patients and for preventing life-threatening illness, disability, and death [2].

In 2007, Sierra Leone established a database for monitoring ADRs known as the VigiFlow. This database is managed by the National Pharmacovigilance Centre which is housed at the Pharmacy Board of Sierra Leone [3]. The data from VigiFlow feeds into the World Health Organization’s (WHO) global database for international drug monitoring known as VigiBase [4]. ADR data once entered into VigiFlow are thereafter referred to as Individual Case Safety Reports (ICSRs).

The VigiFlow database is intended to gather ICSRs from a country-wide perspective and generate information that could guide drug safety, drug regulation systems, and clinical practicein Sierra Leone. Fawzi et al. [5], through the Structured and Operational Research and Training IniTiative (SORT-IT), revealed shortcomings in the timeliness and completeness of ADR reports. Ninety percent of reports crossed the 30-day time limit for reporting, 58% of reports were incomplete and 36% of patient outcomes were classified as ‘recovering’, with no final ascertained patient outcome available [5].

These findings were disseminated and served as a ‘wake-up call’ to decision makers. This led to a number of actions to improve the quality of reporting: updating standard operating procedures and guidelines; setting performance targets for monitoring; introducing a tracking system to ensure timely reporting; ascertaining ADR outcomes; and training healthcare workers on various aspects of pharmacovigilance.

A PUBMED search revealed no previous studies had assessed the impact (before-and-after) of introducing quality control interventions to improve the reporting of ADRs. 

We thus aimed to describe the dissemination activities, recommendations, and actions taken to improve the quality of ICSRs and their impact. For assessing the impact on the quality of reporting, the study by Fawzi et al. [5] was considered the baseline study and the current study, the follow-up study. Henceforth they are referred to as such.

The specific objectives of this study were to compare country-wide reporting of ICSRs for a period before (January 2017–December 2021) and after (June 2022–April 2023) the introduction of interventions to improve the quality of reporting. The parameters that were compared included (a) timeliness and completeness of reporting and (b) proportion of patient outcomes classified as still ‘recovering’. 

## 2. Materials and Methods

### 2.1. Study Design

A before-and-after comparative study of ICSRs routinely entered into VigiFlow.

### 2.2. General Setting

Sierra Leone is located in West Africa and has about 8 million inhabitants, and the capital city is Freetown [6]. The country is divided into five administrative regions: the Northern Region, North-Western Region, Eastern Region, Southern Region, and the Western Area. The five regions are sub-divided into 16 districts. 

### 2.3. The National Pharmacovigilance Center and ADR Reporting 

Details of the National Pharmacovigilance Center and ADR reporting have been described in the baseline study [5]. In brief, all 29 public hospitals in Sierra Leone are pharmacovigilance reporting sites and have a pharmacovigilance focal point [3,7]. Reporting is performed through four channels: (a) a healthcare provider fills out the ADR report; (b) one of the designated pharmacovigilance focal persons fills out the report; (c) any individual in the community can access a web link for reporting using an electronic ADR form by using their phones or computers; and (d) an application (MedSafetyApp, Version 24.1.6.24106) can be downloaded and used by anyone to fill out the ADR report. The latter was introduced in December 2022. 

### 2.4. Processing of Individual Case Safety Reports (ICSRs) in VigiFlow

The processing of ICSRs in VigiFlow has been described previously [5]. The VigiFlow includes ICSRs from all kinds of commonly used drugs such as antimicrobials, drugs used for non-communicable diseases (hypertension, diabetes, etc.), skin infections, and mass drug campaigns. Following the quality control interventions introduced to improve reporting quality, all ADR forms are now systematically cross-checked by central pharmacovigilance staff and cross-validated before and after entry into VigiFlow. A completeness check is performed to verify whether the 11 required variables have been filled in—these variables include: identification, sex, age, name of suspected drug, strength, dose, the start date of administration, description of the reaction, therapeutic indication, onset time of the reaction, outcome, and reporters’ details. If any of these variables are missing, a follow-up inquiry is made by the pharmacovigilance staff. During the period of the follow-up study (June 2022–April 2023), ADR outcomes were routinely ascertained to improve the classification of final patient outcomes.

### 2.5. Dissemination Activities, Recommendations, and Actions Taken 

Following the baseline study published in 2022 [5], a specific SORT IT module was conducted in May 2022 to develop the practical skills and tools to effectively communicate the research findings [8]. These tools included: (1) a communication plan to target decision-makers and stakeholders; (2) a one-page plain language summary of the key messages written in a short and simple manner; (3) PowerPoint presentations to be used at national fora and conferences, and (4) an elevator pitch—a one-minute oral presentation for one-to-one conversations with decision makers [8]. Table 1 shows the dissemination details including how dissemination was carried out, to whom and when, and the number of individuals targeted [8].

Table 2 shows the recommendations, action status, and details of actions for improving ADR reporting. This information was sourced from the published study and the plain language summary [8], and was complemented by the study team of the baseline study who are also co-authors of the current study. 

The principal investigator of the baseline study [5] provided detailed information on the efforts made to disseminate the results to the key stakeholders and the actions taken.

### 2.6. Study Population and Period

All ICSRS available in the pharmacovigilance database (VigiFlow) for the period January 2017–December 2021 (baseline study) and June 2022 to April 2023 (follow-up study).

### 2.7. Sample Size Calculation

The baseline study that assessed the quality of reporting between January 2017 and December 2021 showed an overall completeness score in ICSR reporting of 43.8% using a sample of 566 consecutive ICSRs (baseline cohort). Following the implementation of interventions to improve the quality of reporting, we estimated an at least 75% improvement in overall completion score—from 43.8% to 75% or above) by 2023.With such a high level of expected improvement, a minimal sample size of 54 patients was required in each group (including the follow-up cohort) toachieve a power of 90%, and to detect a 75% absolute improvement in completion score with a type I error of 5%. This calculation formed the basis of using a 10-month minimum recruitment period of consecutive patients for the follow-up cohort.

### 2.8. Data Sources, Collection and Validation

Data variables related to dissemination were sourced from the communication plans and a listof stakeholders who attended various dissemination events. Data on ICSRs was extracted from the VigiFlow database. The variables included patient identifiers, age, gender, date of onset of ADR symptoms, seriousness, date of entry into VigiFlow, and patient outcomes. ICSR data were cross-validated as part of the routine pharmacovigilance procedures before entry into VigiFlow.

### 2.9. Statistical Analysis

The ICSR data were exported to Microsoft Excel (Version 10, 2018) and analyzed using STATA (StatCorp, College Station, TX, USA). For the purposes of this study, all suspected ADRs were considered as ADRs. The time taken to report an ADR (ADR reporting time) was calculated as the time from the date of onset of reactions to the date of entry into the VigiFlow database. For each ICSR, the completeness score is auto-calculated by the VigiFlow based on the completeness in reporting the required variables (the calculated score lies between 0 and 100%). To assess the timeliness of reporting, we compared the proportion of ICSR reports received within 30 days in the before-and-after study periods. Similarly, for completeness, we compared the proportion of reports with ≥90% completeness in variables of ICSRs entered in VigiFlow. The ≥90% completeness threshold was an initial arbitrary level used for the purposes of this study.

Frequencies and proportions were used to report on patient outcomes. The chi-squared test was used to compare differences in proportions. The level of confidence was set at 95% and a *p*-value of ≤0.05 was considered significant.

## 3. Results

### 3.1. Timeliness and Completeness of Reporting

Table 3 shows the timeliness and completeness of reporting of ICSRs in the baseline and follow-up operational research studies. There were 566 ICSRs reported during the baseline study and 59 during the follow-up.

In the baseline study, 10%of ICSRs were reported within the stipulated time limit of <30 days while in the follow-up study this increased to 47% (an almost five-fold improvement, *p* < 0.001).

In terms of completeness of reporting (≥90%), 44% ICSRs were complete at baseline compared to 80% in the follow-up study (a 1.8-fold improvement). During the baseline study, 10% of all ICSRs had few variables entered and were ineligible for the calculation of completeness scores. During the follow-up period, all ICSRs were eligible and assessed for completeness scores.

The proportion of ICSRs reported by Community Health Officers increased significantly from 3% (baseline study) to 40% (follow-up study, *p* < 0.001).

### 3.2. Ascertainment of Patient Outcomes

Table 4 shows patient outcomes for antimicrobial-related ADRs in the baseline and follow-up operational research studies. In the baseline study, 36% of patient outcomes were classified as recovering, with no ascertained final patient outcome. In the follow-up study, this significantly decreased to 3% (*p* < 0.001), with all outcomes ascertained. There were a total of four deaths including both the baseline and follow-up periods. None of these deaths were due to ADRs. In three individuals, the cause of death was severe malaria, and one had severe pneumonia.

## 4. Discussion

This before-and-after operational research study showed that following the introduction of quality control interventions, there was a five-fold improvement in timeliness and two-fold improvement in the completeness of ICSRs. There was also a significant reduction in patient outcomes classified as ‘recovering’, implying better routine asan ascertainmentof final outcomes.

This study is important as it shows the role operational research can play in informing decisions and actions for improving the pharmacovigilance system which would provide a more reliable picture of the safety profiles of medicines being used in the country. In particular, timely monitoring of serious ADRs is vital to guide the safer use of medicines in health facilities and in the community at large. Robust pharmacovigilance data is also key to guiding drug regulation systems, clinical practice, and programming, and these findings show that Sierra Leone is on the right track towards achieving this goal. 

The study’s strengths are that (a) we used country-wide data and the same parameters to assess quality during the baseline and follow-up studies; (b) the study theme is anational operational research priority and thus policy relevant, and (c) the relatively short period between the baseline and follow-up assessments reduces the likelihood that factors other than the main intervention (quality control interventions) would have come into play. We also adhered to STROBE guidelines for the reporting of observational studies in epidemiology [9].

A limitation of the studyis that 8%of ICSRs included inthe follow-up period originated from a time when quality control interventions were in the process of being fully implemented and health facilities were getting used to the new procedures. This might have negated the overall impactassociation seen in the follow-up period when compared to the baseline. As time goes on, this effect should dampen out, giving a more robust reflection of the associated impact. Another limitation is that we had smaller numbers (59 ICSRs) in the relatively shorter follow-up period (almost one year) compared to 566 in the baseline period (five years) [5]. Nevertheless, the improvements in the follow-up period were highly significant, implying that a strong effect was taking place. Following the various interventions put in place to improve the quality of reporting, we expected a significant improvement of 75% or more in the completeness score of the follow-up cohort compared to the baseline cohort. A minimum of 54 ICSRs were needed in the follow-up cohort which was achieved. Thus, the sample size difference is not of major concern as expected differences in the quality of reporting between the two cohorts were large. Notably, the total ADR reports for antimicrobials used outside mass ivermectin campaigns during 2017 to 2021 was only 52. In Sierra Leone and many other countries, ADR reporting is a passive and “voluntary” activity and this could reflect the low numbers [5]. Although we wished to have a longer period for the follow-up study, this was not possible due to funding constraints. That notwithstanding, we consider this a preliminary evaluation while further operational research is envisaged with a longer follow-up and sample sizes.

The findings from this study have a number of policy and practice implications. First, significant actions were taken in a relatively short time after the publication of the baseline study in March 2022 [5]. By April 2022, (a month after publication) the study findings were communicated to stake holders and by June 2022 (three months later), five major quality control interventions recommended by the baseline study were put in place. These included: (a) training; (b) updatingstandard operating procedures and guidelines; (c) setting performance thresholds for reporting; (d) introducing a tracking system to monitor and enforce timely reporting and data entry into VigiFlow and; (e) optimizing the use of MedSafety application for ADR reporting using mobile phones. As there were no other changes or interventions made to pharmacovigilance reporting in the country, we believe that the observed improvements are directly associated with these interventions.

The enabling factors for such a rapid uptake of research findings would include: research relevance; early involvement of decision makers and their engagement as study authors, therebyenhancing co-ownership and responsibility; embedding research within the routine health system, and effectively disseminating research findings. Six of the co-authors were also influential decision makers who shape policy and practice on pharmacovigilance in Sierra Leone. In principle, we encouraged ‘local research, with local ownership, for local solutions’. This experience highlights the importance of engaging throughout the research process with decision makers [10,11]. A PubMed search found studies from Nigeria [12], Ghana [13], Togo [14], and Zimbabwe [15] that assessed ADRs at a country-widelevel and showed shortcomings in the quality of reporting, but unlike Sierra Leone none of these studies performed a reassessment to assess if there were improvements thereafter.

Although we continue to make efforts towards making further improvements, the improvements that have been made in Sierra Leone are significant while the country still faces teething problems in establishing robust ADR reporting systems as in other settings [14,15,16].

Second, in the baseline study, 36% of individuals with ADRs were classified as still ‘recovering’ even when some of the reported ADRs dated as far back as two years prior to analysis. This implied that routine ascertainment of patient outcomes was not occurring. In the follow-up period, a significant improvement was seen with a decrease in the proportion of those ‘recovering’ to 2%. This can be explained by the fact that health workers were trained on the importance of updating preliminary ADR reports once final ADR patient outcomeswere ascertained. The national pharmacovigilance team also routinely followed up on patient outcomes. Such ascertainment of final patient outcomes (e.g., deaths) is vital to judging medicine safety.

Finally, the proportion of ADRs reported by Community Health Officers who work in peripheral health facilities increased from 3% to 40%. Clinical trial sites also came on board for ADR reporting, which is encouraging and contributes to better pharmacovigilance. This is important in ensuring decentralized patient safety and changes the traditional perspective that pharmacovigilance is a pharmacist-driven activity. It is notable that the contribution by consumers and non-health professionals was two records in the baseline cohort and none in the follow-up cohort. This is an area that needs focused attention.

## 5. Conclusions

In conclusion, following a wake-up callfor improving the ADR reporting system inSierra Leone, significant improvements in timeliness, completeness, and validation of ICSRs were observed. While enhancing pharmacovigilance and patient safety, this study also highlights the important synergistic role operational research can play in improving monitoring and evaluation systems in the country.

## Figures and Tables

**Table 1 tropicalmed-08-00470-t001:** Dissemination details of the baseline operational research study (2022), Sierra Leone [5].

How	To Whom	Where (Number) ^1^	When
Stakeholder mapping and communication planning	Study team	SORT IT workshop venue	April 2021
Lightening presentations and discussions	Key stakeholders of the One Health platform; WHO office, Sierra Leone	Atlantic Lumley hotel, Freetown (50)	May 2022
Publication in a peer-reviewed journal	Pharmacovigilance forum for healthcare professionals and forum for Pharmacy professionals	Ref. [5] (2130 article views) Whatsapp group (508)	June 2022
Evidence summaries	Key stakeholders of the One Health platform WHO office, Sierra Leone Pharmacovigilance forum for healthcare professionals Hospital and district health management teams ^2^	Various locations (356)	June 2022
PowerPoint presentation and discussions	Pharmacy Board of Sierra Leone	Central Medical Stores, New England Ville (12)	June 2022
	Hospital and district health management teams	Bo and Kenema government hospitals (40)	November 2022
	Ministry of health stakeholders World Health Organization	Sierra palms hotel, Freetown (50)	November 2022
	Community pharmacy professionals	Pharmacy board of Sierra Leone (20)	December 2022
Publication uploaded on websites	National and international stakeholders/general public	Pharmacy Board website TDR website	December 2022

^1^ Number of individuals attending the dissemination activity/event. ^2^ Hospital and district health management teams: includes district medical officers, public health program officers, district pharmacists, logistic officers, nurses and nurse assistants, monitoring and evaluation officers, and community health officers. Abbreviations: SORT IT—Structured Operational Research Training Initiative; TDR—the Special Programme for Research and Training in Tropical Diseases; WHO—World Health Organization.

**Table 2 tropicalmed-08-00470-t002:** Recommendations, action status and details of action from the baseline study (2022) for improving Adverse Drug Reaction reporting [5].

Recommendation	Action Status	Details of Action (When and What)
Updating standard operating procedures and guidelines	Implemented	*June 2022* Parameters for ensuring completeness and timeliness of data were introduced
Introduction of performance targets for reporting ADRs	Implemented	*June 2022* Serious ADRs to be reported within 7 days and others within 30 days All 11 key variables on ADR forms to be filled in Systematic cross-validation of data entered into VigiFlow by central-level staff A checklist for mandatory variables updated and a follow-up system (using phone calls, text messages, emails) put in place to gather missing information
Introduction of a tracking system to ensure timely reporting and data entry into VigiFlow	Implemented	*June 2022* Tracker for processing of ADR reports in place
Optimization of the online electronic ADR reporting system using mobile phones and/or computers.	Implemented	*June 2022* Online platform was activated.
Introduction of the MedSafety version XXapplication for data capture using mobile phones/computers	Implemented	*December 2022* Medsafety application launched
Compulsory ADR reporting at all health facilities	Partially implemented and ongoing	*June 2012–2023* Training and awareness raising on ADR reporting by District Health Management teams including community health officers.

Abbreviations: ICSR—Individual Case Safety Report; ADR—Adverse drug reaction.

**Table 3 tropicalmed-08-00470-t003:** Timeliness and completeness of reporting of individual case safety reports (ICSR) to the VigiFlow, Sierra Leone in a baseline (January 2017–December 2021) and follow-up operational research studies (June 2022–April 2023).

Characteristics	Baseline Study		Follow-Up Study		*p*-Value ^1^
	n	(%) *	n	(%) *	
**Total**	566	(100)	59	(100)	
**Time to reporting (in days) ^2^**					
<30	56	(9.9)	28	(47.4)	<0.001
30–180	289	(51.1)	17	(28.8)	0.001
≥180	221	(39.0)	14	(23.7)	0.019

**Completeness score ^3^**					
≥90%	248	(43.8)	47	(79.7)	<0.001
<90%	372	(46.6)	12	(20.3)	<0.001
Not recorded	54	(9.5)	0	(0)	-

**Reported by**					
Pharmacists	364	(64.3)	26	(44.1)	0.002
Physicians and nurses	132	(23.3)	9	(15.3)	0.2
Consumers or non-health professionals	2	(0.4)	0	(0)	-
Community health officers	15	(2.7)	24	(40.1)	<0.001
Not recorded	53	(9.4)	0	(0)	-

* Column percentage. ^1^ Chi-Square test; ^2^ Duration between the start of ADR (symptom onset) and entry into VigiFlow; ^3^ Completeness score is auto-calculated by the VigiFlow for each ICSR based on the proportion of variablescorrectly filled into the ADR form (score lies between 0 to 100%). Abbreviations: ICSR—Individual Case Safety Report; ADR—adverse drug reaction; OR—Operational Research.

**Table 4 tropicalmed-08-00470-t004:** Patient outcomes for antimicrobial-related Adverse Drug Reactions in individual case safety reports in the VigiFlow in Sierra Leone, in a baseline (January 2017–December 2021) and follow-up operational research studies (June 2022–April 2023).

Reported Patient Outcome	Base Line Study		Follow-Up Study		*p*-Value ^1^
	n	(%)	n	(%)	
**Total ^2^**	**566**		**59**		
Recovered	337	(59.5)	50	(84.7)	<0.001
Recovering	201	(35.5)	2	(3.3)	<0.001
Not recovered	8	(1.4)	1	(1.7)	0.6
Death	1	(0.2)	3	(5.1)	<0.001
Unknown	19	(3.4)	3	(5.1)	0.7

^1^ Chi-Square test. ^2^ Recovered = adverse drug reaction has stopped; Recovering = adverse drug reaction is gradually reducing; Not recovered = reaction persists; Death = loss of life; Unknown = no knowledge on patient outcome.

## Data Availability

Request to access these data should be sent to the corresponding author.

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
