# Peer review of "Quality of Reporting of Adverse Drug Reactions to Antimicrobials Improved Following Operational Research: A before-and-after Study in Sierra Leone (2017–2023)"

_tropicalmed, 2023, doi:10.3390/tropicalmed8100470_

Round 1

Reviewer 1 Report

The manuscript aims to improve the quality of reporting of Individual Case Reports in Sierra Leone.

The main focus of the manuscript is of real interest to readers, unfortunately the presentation of the data is not sound. Even though the authors state in the discussions section that a limitation of the study may be that the follow up group is significantly smaller that the group analyzed for the baseline period, regardless of the period of time, I consider that this decreases the credibility of the research. I encourage the authors to use more data for the follow up and thus to compare similar numbers of data.

I consider that 11 references are not enough for achieving a quality manuscript. I advise the authors to study the scientific literature and insert more relevant papers to ensure an increase of the quality of this manuscript.

Author Response

Please see the attachment below for your attention.

Thank you

Reviewer 2 Report

All ICSRs or just those related to antimicrobials?

It's unclear why there's a need to focus on antimicrobials only. 59 ISCRs in total but also 59 antimicrobial ISCRs?

- it'll be useful to provide some details and summary statistics of what the drugs in the Vigiflow database. 

The slew of interventions to improve ADR reporting had occurred at various timepoints even after June 2022.. is it still correct to assign June 2022 as the appropriate starting point as the 'post' period?  

Line 222 - We also adhered to STROBE guidelines for the reporting of observational studies in epidemiology[9]. 

- Please supply the checklist to demonstrate adherence to this guideline. Althought for this study, it's unclear how much STROBE is relevant here.

Lines 250 to 252 are appropriate as it's not possible to ascribe causation in an observational study however language used in lines 232 and 233 suggest a causal link which is inappropriate. 

I laud the authors for their efforts in improving ADR reporting in their country. I would however mention that just a mere increase in no of reports within a certain time is probably insufficent to protect public health although it is important - there needs to be a culture of safety and reporting which admittedly is very challenging to develop. 

Sometimes efforts to increase reporting can also be counterproductive - when reporting is mandatory, this can lead to false positives (or cases of reporting for the sake of reporting) to avoid punitive action. 

It would be nice to describe some of the regulatory action or communications that the national PV centre has published or announced to healthcare practitioners to better illustrate how increases in reporting have led to other meaningful advantages beyond just an increase in number of reports in the database. 

-

Author Response

(The authors gave the same response as above.)

Reviewer 3 Report

This manuscript refers some interventions to improve the quality of reporting ADRs.

The measures applied lead to an improvement in the reporting. This work can be of use as a reference to make in place measures for other healthcare professionals.

There are some things that I would like to point out:

-       Material and methods: line 101, the authors say that one of the objectives for improvement is the completeness of the 11 variables. It would be interesting to know what are the variables that needed more improvement. Where are the main difficulties when completing the reports.

-       In the manuscript the authors talk about the consumers as reporters of ADRs, but in the actions that they implemented, I understand, that they were not included. Therefore, I consider that the reports derived from consumers should not be included in Table 3.

Author Response

Please see the attachment below for your attention.

Thank you.

Round 2

Reviewer 1 Report

The topic of the manuscript is of real interest. It is very important that reporting should be encouraged. I stand by my conclusion that a correct estimation could not be performed on such different time periods in terms of length, thus such a large difference between the number of reports between the two cohorts.